# Escaping the Big Data Paradigm with Compact Transformers

## Abstract

With the rise of Transformers as the standard for language processing, and their advancements in computer vision, there has been a corresponding growth in parameter size and amounts of training data. Many have come to believe that because of this, transformers are not suitable for small sets of data. This trend leads to concerns such as: limited availability of data in certain scientific domains and the exclusion of those with limited resource from research in the field. In this paper, we aim to present an approach for small-scale learning by introducing Compact Transformers. We show for the first time that with the right size, convolutional tokenization, transformers can avoid overfitting and outperform state-of-the-art CNNs on small datasets. Our models are flexible in terms of model size, and can have as little as 0.28M parameters while achieving competitive results. Our best model can reach 98% accuracy when training from scratch on CIFAR-10 with only 3.7M parameters, which is a significant improvement in data-efficiency over previous Transformer based models being over 10x smaller than other transformers and is 15% the size of ResNet50 while achieving similar performance. CCT also outperforms many modern CNN based approaches, and even some recent NAS-based approaches. Additionally, we obtain a new SOTA result on Flowers-102 with 99.76% top-1 accuracy, and improve upon the existing baseline on ImageNet (82.71% accuracy with 29% as many parameters as ViT), as well as NLP tasks. Our simple and compact design for transformers makes them more feasible to study for those with limited computing resources and/or dealing with small datasets, while extending existing research efforts in data efficient transformers.

## 1 Introduction

Convolutional neural networks (CNNs) LeCun et al. (1989) have been the standard for computer vision, since the success of AlexNet Krizhevsky et al. (2012). Krizhevsky *et al.* showed that convolutions are adept at vision based problems due to their invariance to spatial translations as well as having low relational inductive bias. He *et al.* He et al. (2016a) extended this work by introducing residual connections, allowing for significantly deeper models to perform efficiently. Convolutions leverage three important concepts that lead to their efficiency: *sparse interaction*, *weight sharing*, and *equivariant representations* Goodfellow et al. (2016). Translational equivariance and invariance are properties of the convolutions and pooling layers, respectively Goodfellow et al. (2016); Schmidhuber (2015). They allow CNNs to leverage natural image statistics and subsequently allow models to have higher sampling efficiency Ruderman & Bialek (1994;?).

On the other end of the spectrum, Transformers have become increasingly popular and a major focus of modern machine learning research. Since the advent of Attention is All You Need Vaswani et al. (2017), the research community saw a spike in transformer-based and attention-based research. While this work originated in natural language processing, these models have been applied to other fields, such as computer vision. Vision Transformer (ViT) Dosovitskiy et al. (2020) was the first major demonstration of a pure transformer backbone being applied to computer vision tasks. ViT highlights not only the power of such models, but also that large-scale training can trump inductive biases. The authors argued that "*Transformers lack some of the inductive biases inherent to CNNs, such as translation equivariance and locality, and therefore do not generalize well when trained on insufficient amounts of data.*" Over the past few years, an explosion

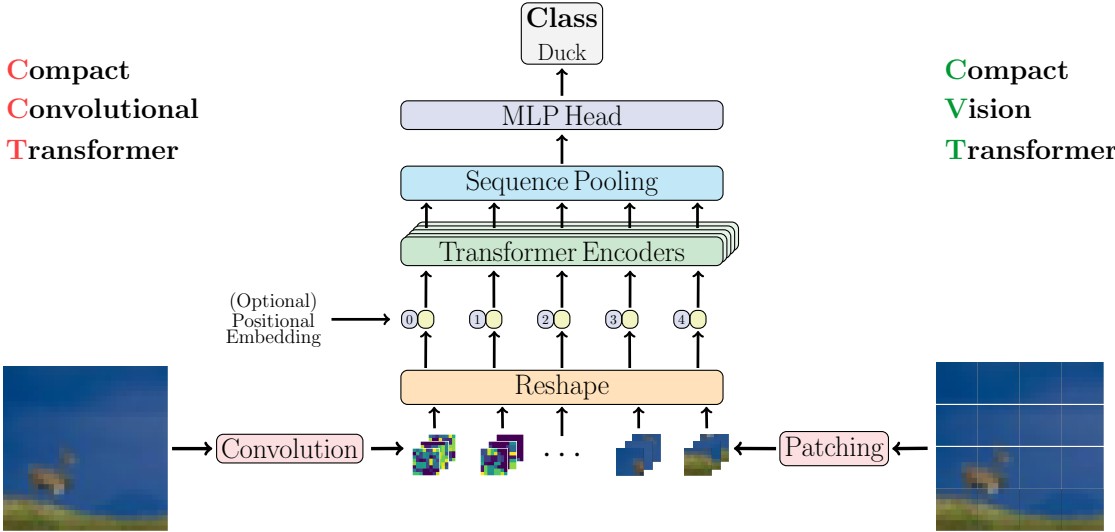

Figure 1: Overview of **CVT** (right), the basic compact transformer, and **CCT** (left), the convolutional variant of our compact transformer. CCT can be quickly trained from scratch on small datasets, while achieving high accuracy (in under 30 minutes one can get 90% on an NVIDIA 2080Ti GPU or 80% on an AMD 5900X CPU on CIFAR-10).

in model sizes and datasets has also become noticeable which has led to a "data hungry" paradigm, making training transformers from scratch seem intractable for many types of pressing problems, where there are typically several orders of magnitude less data. It also limits major contributions in the research to those with vast computational resources. As a result, CNNs are still the go-to models for smaller datasets because they are more efficient, both computationally and in terms of memory, when compared to transformers. Additionally, local inductive bias shows to be more important in smaller images. They require less time and data to train while also requiring a lower number of parameters to accurately fit data. However, they do not enjoy the long range interdependence that attention mechanisms in transformers provide. Reducing machine learning's dependence on large sums of data is important, as many domains, such as science and medicine, would hardly have datasets the size of ImageNet Deng et al. (2009). This is because events are far more rare and it would be more difficult to properly assign labels, let alone create a set of data which has low bias and is appropriate for conventional neural networks. In medical research, for instance, it may be difficult to compile positive samples of images for a rare disease without other correlating factors, such as medical equipment being attached to patients who are actively being treated. Additionally, for a sufficiently rare disease there may only be a few thousand images for positive samples, which is typically not enough to train a network with good statistical prediction unless it can sufficiently be pre-trained on data with similar attributes. This inability to handle smaller datasets has impacted the scientific community where they are much more limited in the models and tools that they are able to explore. Frequently, problems in scientific domains have little in common with domains of pre-trained models and when domains are sufficiently distinct pre-training can have little to no effect on the performance within a new domain Zhuang et al. (2020). In addition, it has been shown that strong performance on ImageNet does not necessarily result in equally strong performance in other domains, such as medicine Ke et al. (2021). Furthermore, the requisite of large data results in a requisite of large computational resources and this prevents many researchers from being able to provide insight. This not only limits the ability to apply models in different domains, but also limits reproducibility. Verification of state of the art machine learning algorithms should not be limited to those with large infrastructures and computational resources. The above concerns motivated our efforts to build more efficient models that can be effective in less data intensive domains and allow for training on datasets that are orders of magnitude smaller than those conventionally seen in computer vision and natural language processing (NLP) problems. Both Transformers and CNNs have highly desirable qualities for statistical inference and prediction, but each comes with their own costs. In this work, we try to bridge the gap between these two architectures and develop an architecture that can both attend to important features

within images, while also being spatially invariant, where we have sparse interactions and weight sharing. This allows for a Transformer based model to be trained from scratch on small datasets like CIFAR-10 and CIFAR-100, providing competitive results with fewer parameters and low computational requirements.

In this paper we introduce ViT-Lite, a smaller and more compact version of ViT, which can obtain over 90% accuracy on CIFAR-10. We expand on ViT-Lite by introducing a sequence pooling and forming the Compact Vision Transformer (CVT). We further iterate by adding convolutional blocks to the tokenization step and thus creating the Compact Convolutional Transformer (CCT). Both of these simple additions add to significant increases in performance, leading to a top-1%accuracy of 98% on CIFAR-10. This makes our work the only transformer based model in the top 25 best performing models on CIFAR-10, without pre-training, and significantly smaller than the vast majority. Our model also outperforms most comparable CNN-based models within this domain, with the exception of certain Neural Architectural Search techniques Cai et al. (2018). Additionally, we show that our model can be lightweight, only needing 0.28 million parameters and still reach close to 90% top-1% accuracy on CIFAR-10. On ImageNet, CCT achieves 80.67% accuracy while still maintaining a small number of parameters and reduced computation. CCT outperforms ViT, while containing less than a third of the number of parameters with about a third of the computational complexity (MACs). Additionally, CCT outperform similarly sized and more recent models, such as DeiT Huang et al. (2020). This demonstrates the scalability of our model while maintaining compactness and computational efficiency. The main contributions of this paper are:

- Extending transformer-based research to small data regimes, by introducing ViT-Lite, which can be trained from scratch and achieve high accuracy on datasets such as CIFAR-10.

- Introducing Compact Vision Transformer (CVT) with a new sequence pooling strategy, which pools over output tokens and improves performance.

- Introducing Compact Convolutional Transformer (CCT) to increase performance and provide flexibility for input image sizes while also demonstrating that these variants do not depend as much on Positional Embedding compared to the rest.

In addition, we demonstrate that our CCT model is fast, obtaining 90% accuracy on CIFAR-10 using a single NVIDIA 2080Ti GPU and 80% when trained on a CPU (AMD 5900X), both in under 30 minutes. Additionally, since our model has a relatively small number of parameters, it can be trained on the majority of GPUs, even if researchers do not have access to top of the line hardware. Through these efforts, we aim to help enable and extend research around Transformers to cases with limited data and/or researchers with limited resources.

## 2    Related Works

In NLP research, attention mechanisms Graves et al. (2014); Bahdanau et al. (2016); Luong et al. (2015) gained popularity for their ability to weigh different features within sequential data. Transformers Vaswani et al. (2017) were introduced as a fully attention-based model, primarily for machine translation and NLP in general. Following this, attention-based models, specifically transformers have been applied to a wide variety of tasks beyond machine translation Devlin et al. (2019); Liu et al. (2019); Yang et al. (2019), including: visual question answering Lu et al. (2019); Su et al. (2019), action recognition Bertasius et al. (2021); Girdhar et al. (2019), and the like. Many researchers also leveraged a combination of attention and convolutions in neural networks for visual tasks Wang et al. (2017); Hu et al. (2018); Bello et al. (2019); Zhang et al. (2019). Ramachandran *et al.* Ramachandran et al. (2019) introduced one of the first vision models that rely primarily on attention. Dosovitskiy *et al.* Dosovitskiy et al. (2020) introduced the first stand-alone transformer based model for image classification (ViT). In the following subsections, we briefly revisit ViT and several other related works.

### 2.1    Vision Transformer

Dosovitskiy *et al.* Dosovitskiy et al. (2020) introduced ViT primarily to show that reliance on CNNs or their structure is unnecessary, as prior to it, most attention-based models for vision were used either with

convolutions Wang et al. (2017); Bello et al. (2019); Zhang et al. (2019); Carion et al. (2020), or kept some of their properties Ramachandran et al. (2019). The motivation, beyond self-attention's many desirable properties for a network, specifically its ability to make long range connections, was scalability. It was shown that ViT can successfully keep scaling, while CNNs start saturating in performance as the number of training samples grew. Through this, they concluded that large-scale training triumphs over the advantage of inductive bias that CNNs have, allowing their model to be competitive with CNN based architectures given sufficiently large amount of training data. ViT is composed of several parts: Image Tokenization, Positional Embedding, Classification Token, the Transformer Encoder, and a Classification Head. These subjects are discussed in more detail below.

**Image Tokenization:** A standard transformer takes as input a sequence of vectors, called tokens. For traditional NLP based transformers, word ordering provides a natural order to sequence the data, but this is not so obvious for images. To tokenize an image, ViT subdivides an image into non-overlapping square patches in raster-scan order. The sequence of patches, $\mathbf{x_p} \in \mathbb{R}^{H \times (P^2 C)}$ with patch size $P$, are flattened into 1D vectors and transformed into latent vectors of dimension $d$. This is equivalent to a convolutional layer with $d$ filters, and $P \times P$ kernel size and stride. This simple patching and embedding method has a few limitations, in particular: loss of information along the boundary regions.

**Positional Embedding:** Positional embedding adds spatial information into the sequence. Since the model does not actually know anything about the spatial relationship between tokens, adding extra information to reflect that can be useful. Typically, this is either a learned embedding or tokens are given weights from two sine waves with high frequencies, which is sufficient for the model to learn that there exists a positional relationship between these tokens.

**Transformer Encoder:** A transformer encoder consists of a series of stacked encoding layers. Each encoder layer is comprised of two sub-layers: Multi-Headed Self-Attention (MHSA) and a Multi-Layer Perceptron (MLP) head. Each sub-layer is preceded by a layer normalization (LN), and followed by a residual connection to the next sub-layer.

**Classification:** Vision transformers typically add an extra learnable `[class]` token to the sequence of the embedded patches, representing the class parameter of an entire image and its state after transformer encoder can be used for classification. `[class]` token contains latent information, and through self-attention accumulates more information about the sequence, which is later used for classification. ViT Dosovitskiy et al. (2020) also explored averaging output tokens instead, but found no significant difference in performance.

## 2.2 Data-Efficient Transformers

In an effort to reduce dependence on data, Touvron *et al.* Touvron et al. (2020) proposed Data-Efficient Image Transformers (DeiT). Using more advanced training techniques, and a novel knowledge transfer method, DeiT improves the classification performance of ViT on ImageNet-1k without large-scale pre-training on datasets such as JFT-300M Sun et al. (2017) or ImageNet-21k Deng et al. (2009). By relying only on more augmentations Cubuk et al. (2020) and training techniques Zhang et al. (2017); Yun et al. (2019), it is shown that much smaller ViT variants that were unexplored by Dosovitskiy *et al.* can outperform the larger ones on ImageNet-1k without pre-training. Furthermore, DeiT variants were pushed even further through their novel knowledge transfer technique, specifically when using a convolutional model as the teacher. This work pushes forward accessibility of transformers in medium-sized datasets, and we aim to follow by extending the study to even smaller sets of data and smaller models. However, we base our work on the notion that *if a small dataset happens to be sufficiently novel, pre-trained models will not help train on that domain* and the model will not be appropriate for that dataset. While knowledge transfer is a strong technique, it requires a pre-trained model for any given dataset, adding to training time and complexity, with an additional forward pass, and as pointed out by Touvron *et al.* is usually only significant when there's a convolutional teacher available to transfer the inductive biases. As a result, it can be argued that if a network utilized just the bare minimum of convolutions, while keeping the pure transformer structure, it may need to rely less on large-scale training and transfer of inductive biases through knowledge transfer.

Yuan *et al.* Yuan et al. (2021b) proposed Tokens-to-token ViT (T2T-ViT), which adopts a window- and attention-based tokenization strategy. Their tokenizer extracts patches of the input feature map, similar

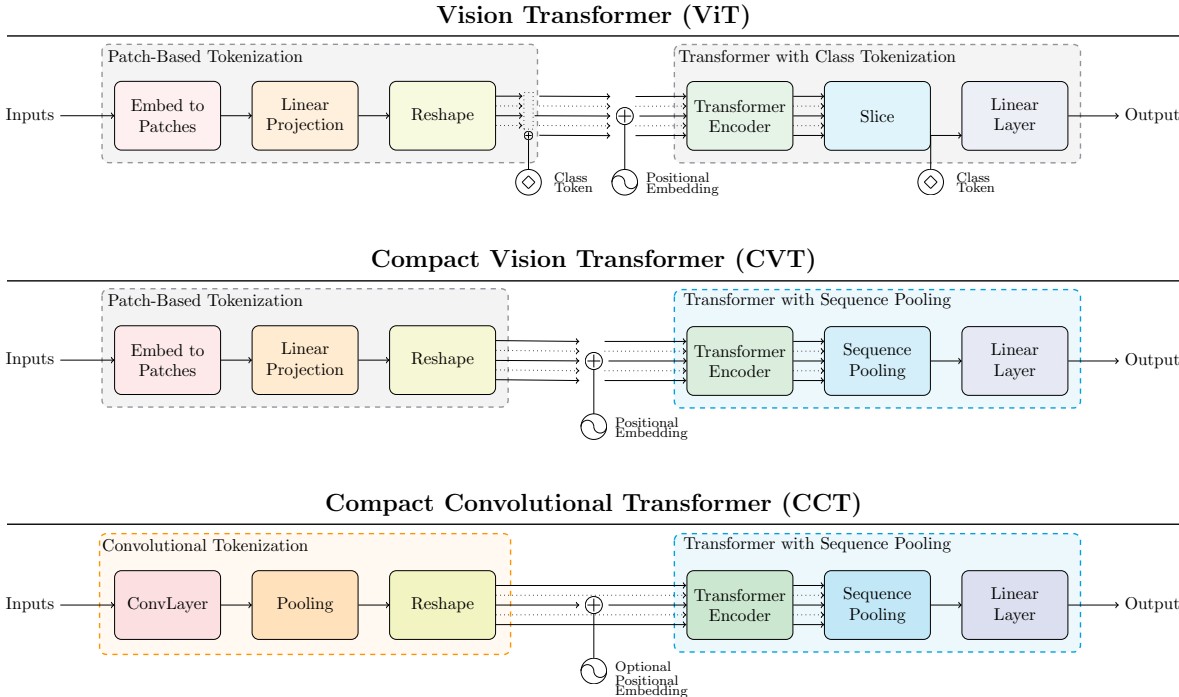

Figure 2: Comparing ViT (top) to CVT (middle) and CCT (bottom). CVT can be thought of as an ablated version of CCT, only utilizing sequence pooling and not a convolutional tokenizer.

to a convolution, applies three sets of kernel weights, and produces three sets of feature maps, which are fed to self-attention as query and key-value pairs. This process is equivalent to convolutions producing the QKV projections in a self-attention module. Finally, this strategy is repeated twice, followed by a final patching and embedding. The entire process replaces patch and embedding in ViT. This strategy, along with their small-strided patch extraction, allows their network to model local structures, including along the boundaries between patches. This attention-based patch interaction leads to finer-grained tokens which allow T2T-ViT to outperform previous Transformer-based models on ImageNet. T2T-ViT differs from our work, in that it focuses on medium-sized datasets like ImageNet, which are not only far too large for many research problems in science and medicine but also resource demanding. T2T tokenizer also has more parameters and complexity compared to a convolutional one.

## 2.3   Hierarchical and  Convolution-inspired Transformers

Hierarchical Vision Transformers, such as Swin Transformer Liu et al. (2021), PiT Heo et al. (2021), PVT Wang et al. (2021), stack multiple transformer encoders with downsampling modules in between, aimed at producing multi-scale feature maps which can be fed to many existing downstream frameworks. These works restrict self attention to linear variants in order to maintain a reasonable memory footprint and complexity, and have at times exceeded existing CNNs in downstream tasks such as object detection and image segmentation.

Many works motivated by Vision Transformers and their performance at scale propose hybrid models made of both attention and convolutions.  ConViT d'Ascoli et al. (2021) introduces a *gated positional self-attention* (GPSA) that allows for a "soft" convolutional inductive bias within their model. GPSA allows their network to have more flexibility with respect to positional information. Since GPSA is able to be initialized as a convolutional layer, this allows their network to sometimes have the properties of convolutions or alternatively having the properties of attention. Its *gating parameter* can be adjusted by the network, allowing it to become more expressive and adapt to the needs of the dataset. Convolution-enhanced image Transformers (Ceit) Yuan et al. (2021a) utilize convolutions throughout their model. They propose a convolution-based

Image-to-Token module for tokenization. They also re-design the encoder with layers of multi-headed self-attention and their novel Locally Enhanced Feedforward Layer, which processes the spatial information form the extracted token. This allows creates a network that is competitive with other works such as DeiT Touvron et al. (2020) on ImageNet. Convolutional vision Transformer (CvT) Wu et al. (2021) introduces convolutional transformer encoder layers, which use convolutions instead of linear projections for the QKV in self-attention. They also introduce convolutions into their tokenization step, and report competitive results compared to other vision transformers on ImageNet-1k. All of these works report results when trained from scratch on ImageNet (or larger datasets).

### 2.4 Comparison

Our work differs from the aforementioned in several ways, in that it focuses on answering the following question: **Can vision transformers be trained from scratch on small datasets?** Focusing on a small datasets, we seek to create a model that can be trained, from scratch, on datasets that are orders of magnitude smaller than ImageNet. Having a model that is compact, small in size, and efficient allows greater accessibility, as training on ImageNet is still a difficult and data intensive task for many researchers. Thus our focus is on an accessible model, with few parameters, that can quickly and efficiently be trained on smaller platforms while still maintaining SOTA results.

## 3 Method

In order to provide empirical evidence that vision transformers are trainable from scratch when dealing with small sets of data, we propose three different models: ViT-Lite, **C**ompact **V**ision **T**ransformers (CVT), and **C**ompact **C**onvolutional **T**ransformers (CCT). ViT-Lite is nearly identical to the original ViT in terms of architecture, but with a more suitable size and patch size for small-scale learning. CVT builds on this by using our **Seq**uence **Pool**ing method (SeqPool), that pools the entire sequence of tokens produced by the transformer encoder. SeqPool replaces the conventional `[class]` token. CCT builds on CVT and utilizes a convolutional tokenizer, generating richer tokens and preserving local information. The convolutional tokenizer is better at encoding relationships between patches compared to the original ViT Dosovitskiy et al. (2020). A detailed modular-level comparison of these models can be viewed in Fig. 2. The components of our compact transformers are further discussed in the following subsections: Transformer-based Backbone, Small and Compact Models, SeqPool, and Convolutional Tokenizer.

### 3.1 Transformer-based Backbone

In terms of model design, we follow the original Vision Transformer Dosovitskiy et al. (2020), and original Transformer Vaswani et al. (2017). As mentioned, the encoder consists of transformer blocks, each including an MHSA layer and an MLP block. The encoder also applies Layer Normalization, $GELU$ activation, and dropout. Positional embeddings can be learnable or sinusoidal, both of which are effective.

### 3.2 Small and Compact Models

We propose smaller and more compact vision transformers. The smallest ViT variant, ViT-Base, includes a 12 layer transformer encoder with 12 attention heads, 64 dimensions per head, and 2048-dimensional hidden layers in the MLP blocks. This, along with the classifier and 16x16 patch and embedder results in over 85M parameters. We propose variants with as few as 2 layers, 2 heads, and 128-dimensional hidden layers. We summarized the details of the variants we propose, the smallest of which can have as little as 0.22M parameters, while the largest (for small-scale learning) only have 3.8M parameters in the appendix. We also adjust the tokenizer (patch size) according to the dataset we're training on, based on its image resolution. These variants, which are mostly similar in architecture to ViT, but different in size, are referred to as ViT-Lite. In our notation, we use the number of layers to specify size, as well as tokenization details: for instance, ViT-Lite-*12*/**16** has *12* transformer encoder layers, and a **16×16** patch size.

### 3.3 SeqPool

In order to map the sequential outputs to a singular class index, ViT Dosovitskiy et al. (2020) and most other common transformer-based classifiers follow BERT Devlin et al. (2019), in forwarding a learnable class or query token through the network and later feeding it to the classifier. Other common practices include global average pooling (averaging over tokens), which have been shown to be preferable in some scenarios. We introduce SeqPool, an attention-based method which pools over the output sequence of tokens. Our motivation is that the output sequence contains relevant information across different parts of the input image, therefore preserving this information can improve performance, and at no additional parameters compared to the learnable token. Additionally, this change slightly decreases computation, due one less token being forwarded. This operation consists of mapping the output sequence using the transformation $T : \mathbb{R}^{b \times n \times d} \mapsto \mathbb{R}^{b \times d}$. Given:

$$\mathbf{x}_L = \mathrm{f}(\mathbf{x}_0) \in \mathbb{R}^{b \times n \times d}$$

where $\mathbf{x}_L$ is the output of an $L$ layer transformer encoder $f$, $b$ is batch size, $n$ is sequence length, and $d$ is the total embedding dimension. $\mathbf{x}_L$ is fed to a linear layer $\mathrm{g}(\mathbf{x}_L) \in \mathbb{R}^{d \times 1}$, and softmax activation is applied to the output:

$$\mathbf{x}'_L = \mathrm{softmax}\left(\mathrm{g}(\mathbf{x}_L)^T\right) \in \mathbb{R}^{b \times 1 \times n}$$

This generates an importance weighting for each input token, which is applied as follows:

$$\mathbf{z} = \mathbf{x}'_L \mathbf{x}_L = \mathrm{softmax}\left(\mathrm{g}(\mathbf{x}_L)^T\right) \times \mathbf{x}_L \in \mathbb{R}^{b \times 1 \times d} \tag{1}$$

By flattening, the output $z \in \mathbb{R}^{b \times d}$ is produced. This output can then be sent through a classifier. SeqPool allows our network to weigh the sequential embeddings of the latent space produced by the transformer encoder and correlate data across the input data. This can be thought of this as attending to the sequential data, where we are assigning importance weights across the sequence of data, only after they have been processed by the encoder. We tested several variations of this pooling method, including learnable and static methods, and found that the learnable pooling performs the best. Static methods, such as global average pooling have already been explored by ViT as well, as pointed out in Related Works. We believe that the learnable weighting is more efficient because each embedded patch does not contain the same amount of entropy. This allows the model to apply weights to tokens with respect to the relevance of their information. Additionally, sequence pooling allows our model to better utilize information across spatially sparse data. We will further study the effects of this pooling in the ablation study. By replacing the conventional `class` token in ViT-Lite with SeqPool, Compact Vision Transformer is created. We use the same notations for this model: for instance, CVT-$7$/$4$ has $7$ transformer encoder layers, and a $4 \times 4$ patch size.

### 3.4 Convolutional Tokenizer

In order to introduce an inductive bias into the model, we replace patch and embedding in ViT-Lite and CVT, with a simple convolutional block. This block follows conventional design, which consists of a single convolution, $ReLU$ activation, and a max pool. Given an image or feature map $\mathbf{x} \in \mathbb{R}^{H \times W \times C}$:

$$\mathbf{x}_0 = \mathrm{MaxPool}(\mathrm{ReLU}(\mathrm{Conv2d}(\mathbf{x}))) \tag{2}$$

where the Conv2d operation has $d$ filters, same number as the embedding dimension of the transformer backbone. Additionally, the convolution and max pool operations can be overlapping, which could increase performance by injecting inductive biases. This allows our model to maintain locally spatial information. Additionally, by using this convolutional block, the models enjoy an added flexibility over models like ViT, by no longer being tied to the input resolution strictly divisible by the pre-set patch size. We seek to use convolutions to embed the image into a latent representation, because we believe that it will be more efficient and produce richer tokens for the transformer. These blocks can be adjusted in terms of downsampling ratio (kernel size, stride and padding), and are repeatable for even further downsampling. Since self-attention has a quadratic time and space complexity with respect to the number of tokens, and number of tokens is equal to the resolution of the input feature map, more downsampling results in fewer tokens which noticeably decreases computation (at the expense of performance). We found that on top of

the added performance gains, this choice in tokenization also gives more flexibility toward removing the positional embedding in the model, as it manages to maintain a very good performance. This is further discussed in Appendix. This convolutional tokenizer, along with SeqPool and the transformer encoder create Compact Convolutional Transformers. We use a similar notation for CCT variants, with the exception of also denoting the number of convolutional layers: for instance, CCT-7/**3x2** has 7 transformer encoder layers, and a 2-layer convolutional tokenizer with **3×3** kernel size.

## 4 Experiments

Table 1: Top-1 validation accuracy comparisons. ⋆ variants were trained longer (see Table IV )

| Model | C-10 | C-100 | Fashion | MNIST | # Params | MACs |
|---|---|---|---|---|---|---|
| *Convolutional Networks (Designed for ImageNet)* | | | | | | |
| **ResNet18** | 90.27% | 66.46% | 94.78% | 99.80% | 11.18 M | 0.04 G |
| **ResNet34** | 90.51% | 66.84% | 94.78% | 99.77% | 21.29 M | 0.08 G |
| **MobileNetV2/0.5** | 84.78% | 56.32% | 93.93% | 99.70% | 0.70 M | < **0.01** G |
| **MobileNetV2/2.0** | 91.02% | 67.44% | 95.26% | 99.75% | 8.72 M | 0.02 G |
| *Convolutional Networks (Designed for CIFAR)* | | | | | | |
| **ResNet56**He et al. (2016a) | 94.63% | 74.81% | 95.25% | 99.27% | 0.85 M | 0.13 G |
| **ResNet110**He et al. (2016a) | 95.08% | 76.63% | 95.32% | 99.28% | 1.73 M | 0.26 G |
| **ResNet1k-v2⋆**He et al. (2016b) | 95.38% | – | – | – | 10.33 M | 1.55 G |
| **Proxyless-G**Cai et al. (2018) | 97.92% | – | – | – | 5.7 M | – |
| *Vision Transformers* | | | | | | |
| **ViT-12/16** | 83.04% | 57.97% | 93.61% | 99.63% | 85.63 M | 0.43 G |
| **ViT-Lite-7/16** | 78.45% | 52.87% | 93.24% | 99.68% | 3.89 M | 0.02 G |
| **ViT-Lite-7/8** | 89.10% | 67.27% | 94.49% | 99.69% | 3.74 M | 0.06 G |
| **ViT-Lite-7/4** | 93.57% | 73.94% | 95.16% | 99.77% | 3.72 M | 0.26 G |
| *Compact Vision Transformers* | | | | | | |
| **CVT-7/8** | 89.79% | 70.11% | 94.50% | 99.70% | 3.74 M | 0.06 G |
| **CVT-7/4** | 94.01% | 76.49% | 95.32% | 99.76% | 3.72 M | 0.25 G |
| *Compact Convolutional Transformers* | | | | | | |
| **CCT-2/3×2** | 89.75% | 66.93% | 94.08% | 99.70% | **0.28** M | 0.04 G |
| **CCT-7/3×2** | 95.04% | 77.72% | 95.16% | 99.76% | 3.85 M | 0.29 G |
| **CCT-7/3×1** | 96.53% | 80.92% | **95.56%** | **99.82%** | 3.76 M | 1.19 G |
| **CCT-7/3×1⋆** | **98.00%** | **82.72%** | – | – | 3.76 M | 1.19 G |

### 4.1 Datasets

We conducted image classification experiments using our method on the following datasets: CIFAR-10, CIFAR-100 (MIT License) Krizhevsky et al. (2009), MNIST, Fashion-MNIST, Oxford Flowers-102 Nilsback & Zisserman (2008) [1], and ImageNet-1k Deng et al. (2009). The first four datasets not only have a small number of training samples, but they are also small in resolution. Additionally, MNIST and Fashion-MNIST only contain a single channel, greatly reducing the information density. Flowers-102 has a relatively small number of samples, while having relatively higher resolution images and 102 classes. We divided these

---

[1]Obtained from Kaggle: https://www.kaggle.com/datasets/nunenuh/pytorch-challenge-flower-dataset

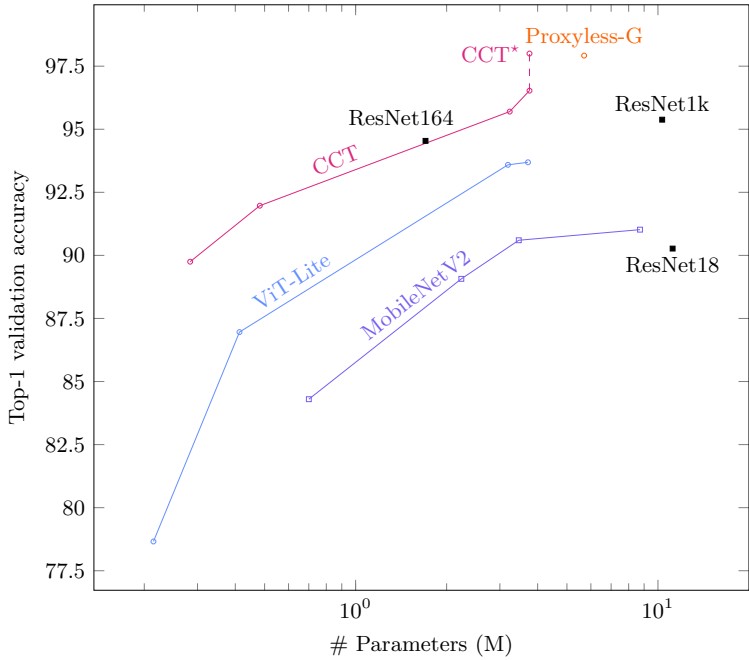

Figure 3: CIFAR-10 accuracy vs model size (sizes < 12M). CCT$^\star$ was trained longer.

Table 2: ImageNet Top-1 validation accuracy comparison (no extra data or pretraining). 🐥 denotes distillation. ResNet50 (2021) is reported from Wightman et al. (2021) which has the same training recipe as ours

| Model | Top-1 | # Params | MACs | Training Epochs |
|---|---|---|---|---|
| **ResNet50** | 77.15% | 25.55 M | 4.15 G | 120 |
| **ResNet50 (2021)** | 79.80% | 25.55 M | 4.15 G | 300 |
| **ViT-S** | 79.85% | **22.05** M | **4.61** G | 300 |
| **CCT-14/7×2** | **80.67%** | 22.36 M | 5.53 G | 300 |
| **DeiT-S** 🐥 | 81.16% | 22.44M | **4.63** G | 300 |
| **CCT-14/7×2** 🐥 | **81.34%** | **22.36** M | 5.53 G | 300 |

datasets into three categories: small-scale small resolution datasets (CIFAR-10/100, MNIST, and Fashion-MNIST), small-scale larger resolution (Flowers-102), and medium-scale (ImageNet-1k) datasets. We also include a study on NLP classification, presented in the appendix.

## 4.2 Performance Comparison

We used the timm package Wightman (2019) to train the models except for cited works which are reported directly. For all CNN experiments, we conducted a hyperparameter sweep for every different method and report the best results we were able to achieve. For Transformer-based models, we only tuned learning rate schedules and set augmentations and other hyperparameters to default values from timm. We will release checkpoints corresponding to the reported numbers, and our hyperparameters in the form of YAML files, along with our code.

**Small-scale small resolution training:** Unless stated otherwise, all tests were run for 200 epochs (10-epoch warmup) with a batch size of 128 and a learning rate of $4e-5$. We follow DeiT Touvron et al. (2020) in adopting the cosine LR scheduler Loshchilov & Hutter (2017), stochastic depth of rate 0.1, label smoothing with a probability of 0.1, and AutoAugment Cubuk et al. (2019) (MIT License), all of which

Table 3: Flowers-102 Top-1 validation accuracy comparison. CCT outperforms other competitive models, having significantly fewer parameters and GMACs. This demonstrates the compactness on small datasets even with large images

| Model | Resolution | Pretraining | Top-1 | # Params | MACs |
|---|---|---|---|---|---|
| **CCT-14/7×2** | 224 | - | 97.19% | 22.17 M | 18.63 G |
| **DeiT-B** | 384 | ImageNet-1k | 98.80% | 86.25 M | 55.68 G |
| **ViT-L/16** | 384 | JFT-300M | 99.74% | 304.71 M | 191.30 G |
| **ViT-H/14** | 384 | JFT-300M | 99.68% | 661.00 M | 504.00 G |
| **CCT-14/7×2** | 384 | ImageNet-1k | **99.76%** | **22.17** M | **18.63** G |

are available in the timm package. In order to demonstrate that vision transformers can be as effective as convolutional neural networks, even in settings with small sets of data, we compare our compact transformers to ResNets He et al. (2016a), which are still very useful CNNs for small to medium amounts of data, as well as to MobileNetV2 Sandler et al. (2018), which are very compact and small-sized CNNs. We also compare with results from He et al. (2016b) where He *et al.* designed very deep (up to 1001 layers) CNNs specifically for CIFAR. The results are presented in Tab. 1, all of which are of models trained from scratch. We highlight the top performers. CCT-7/3x2 achieves on par results with the CNN models, while having significantly fewer parameters in some cases. We also compare our method to the original ViT Dosovitskiy et al. (2020) in order to express the effectiveness of smaller sized backbones, convolutional layers, as well our pooling technique. As these datasets were not trained from scratch in the original paper, we attempted to train the smallest variant: ViT-B/16 (ViT-12/16). We trained our best performing model, CCT-7/3x1, for longer than the 300 epochs to see how far it can go. Surprisingly, this model can get as high as 98% accuracy on CIFAR-10, and 82.87% accuracy on CIFAR-100 when trained for 5000 epochs, which is still fewer iterations an ImageNet pre-training would have. We present results from training on CIFAR-10/100 for 300, 1500 and 5000 epochs in Tab. IV. We observed that sinusoidal positional embedding had a small but noticeable edge over learnable when training longer. This represents the only transformer based model in the top 25 results on PapersWithCode for CIFAR-10 where models have no extra data or pre-training[2]. In addition to this, it is also one of the smallest models, being 15% the size of ResNet50 while maintaining similar performance.

**Medium-scale training:** ImageNet training results are presented in Tab. 2, and compared to ResNet50 He et al. (2016a), ViT, and DeiT. We report ResNet50 from the original paper He et al. (2016a), as well as from Wightman *et al.* Wightman et al. (2021) which uses a similar training schedule to ours, and is therefore a fairer comparison. We also report a smaller ViT variant as proposed by Touvron *et al.* Touvron et al. (2020). We also report CCT's performance with knowledge distillation, in order to compare it to DeiT Touvron et al. (2020). Similar to DeiT, we trained our CCT-14/7x2 with a convolutional teacher and hard distillation loss. We used a RegNetY-16GF Radosavovic et al. (2020) (84M parameters), the same model DeiT selected as the teacher. It is noticeable that distillation does not have as significant of an effect on CCT it does on DeiT. This can be attributed to the already existing inductive biases from the convolutional tokenizer. DeiT authors argued that a convolutional teacher would be able to transfer inductive biases to the student model.

**Small-scale higher-resolution training:** We also present our results on Flowers-102, in which we successfully reach reasonable performance without any pre-training, and with the same model size as our ImageNet model. We also claim state of the art with **99.76%** top-accuracy with ImageNet pretraining, which exceeds even far larger models pre-trained on JFT-300M. In addition to this we note that our model is at least a quarter the size of the next best model and almost 30× smaller than ViT-H/14. CCT is also $3-27\times$ more computationally efficient.

### 4.3 Ablation Study

We present our ablation on model architecture in Tab. 4. We provide a full list of ablated terms showing which factors give the largest boost in performances. "Model" column refers to variant (see Tab. 1 for

---

[2]https://paperswithcode.com/sota/image-classification-on-cifar-10

details), "Conv" specifies the number of convolutional blocks (if an), and "Conv Size" specifies the kernel size. "Aug" denotes the use of AutoAugment Cubuk et al. (2019). "Tuning" specifies a minor change in dropout, attention dropout, and/or stochastic depth (see Tab. 5). The first row in Tab. 4 is essentially ViT. The next three rows are modified variants of ViT, which are not proposed in the original paper. These variants are more compact and use smaller patch sizes. It should be noted that the numbers reported in this table are best out of 4.

Table 4: CIFAR Top-1 validation accuracy when transforming ViT into CCT step by step

| Model | CLS | # Conv | Conv Size | Aug | Tuning | C-10 | C-100 | # Params | MACs |
|---|---|---|---|---|---|---|---|---|---|
| ViT-12/16 | CT | ✗ | ✗ | ✗ | ✗ | 69.82% | 40.57% | 85.63 M | 0.43 G |
| ViT-12/16 | CT | ✗ | ✗ | ✓ | ✓ | 80.72% | 56.73% | 85.63 M | 0.43 G |
| CVT-12/16 | SP | ✗ | ✗ | ✓ | ✓ | 80.84% | 58.05% | 85.63 M | 0.34 G |
| ViT-12/8 | CT | ✗ | ✗ | ✓ | ✓ | 90.24% | 69.81% | 85.20 M | 1.45 G |
| ViT-12/4 | CT | ✗ | ✗ | ✓ | ✓ | 94.07% | 76.08% | 85.12 M | 5.61 G |
| CCT-12/7×1 | SP | 1 | 7 × 7 | ✓ | ✓ | 93.72% | 76.21% | 85.20 M | 5.55 G |
| CCT-12/3×2 | SP | 2 | 3 × 3 | ✓ | ✓ | **94.50%** | **77.05%** | 85.53 M | 5.63 G |
| ViT-Lite-7/16 | CT | ✗ | ✗ | ✗ | ✗ | 71.78% | 41.59% | 3.89 M | 0.02 G |
| ViT-Lite-7/8 | CT | ✗ | ✗ | ✗ | ✗ | 83.38% | 55.69% | 3.74 M | 0.06 G |
| ViT-Lite-7/4 | CT | ✗ | ✗ | ✗ | ✗ | 83.59% | 58.43% | 3.72 M | 0.26 G |
| CVT-7/16 | SP | ✗ | ✗ | ✗ | ✗ | 72.26% | 42.37% | 3.89 M | 0.02 G |
| CVT-7/8 | SP | ✗ | ✗ | ✗ | ✗ | 84.24% | 55.49% | 3.74 M | 0.06 G |
| CVT-7/8 | SP | ✗ | ✗ | ✓ | ✗ | 87.15% | 63.14% | 3.74 M | 0.06 G |
| CVT-7/4 | SP | ✗ | ✗ | ✗ | ✗ | 88.06% | 62.06% | 3.72 M | 0.25 G |
| CVT-7/4 | SP | ✗ | ✗ | ✓ | ✗ | 91.72% | 69.59% | 3.72 M | 0.25 G |
| CVT-7/4 | SP | ✗ | ✗ | ✓ | ✓ | 92.43% | 73.01% | 3.72 M | 0.25 G |
| CVT-7/2 | SP | ✗ | ✗ | ✗ | ✗ | 84.80% | 57.98% | 3.76 M | 1.18 G |
| CCT-7/7×1 | SP | 1 | 7 × 7 | ✗ | ✗ | 87.81% | 62.83% | 3.74 M | 0.26 G |
| CCT-7/7×1 | SP | 1 | 7 × 7 | ✓ | ✗ | 91.85% | 69.43% | 3.74 M | 0.26 G |
| CCT-7/7×1 | CT | 1 | 7 × 7 | ✓ | ✓ | 91.67% | 72.07% | 3.74 M | 0.26 G |
| CCT-7/7×1 | SP | 1 | 7 × 7 | ✓ | ✓ | 92.29% | 72.46% | 3.74 M | 0.26 G |
| CCT-7/3×2 | CT | 2 | 3 × 3 | ✓ | ✓ | 93.36% | 74.77% | 3.85 M | 0.29 G |
| CCT-7/3×2 | SP | 2 | 3 × 3 | ✓ | ✓ | 93.65% | 74.77% | 3.85 M | 0.29 G |
| CCT-7/3×1 | SP | 1 | 3 × 3 | ✓ | ✓ | **94.47%** | **75.59%** | 3.76 M | 1.19 G |

Table 5: Difference between **tuned** and not tuned runs in Table 4.

| Hyper Param | Not Tuned | Tuned |
|---|---|---|
| **MLP Dropout** | 0.1 | 0 |
| **MSA Dropout** | 0 | 0.1 |
| **Stochastic Depth** | 0 | 0.1 |

## 4.4 Performance vs Dataset Size

In this experiment, we evaluated model performance on smaller subsets of CIFAR-10 to determine the relationship between performance and the number of samples within a dataset. Samples were removed

uniformly from each class in CIFAR-10. For this experiment, we compared ViT-Lite and CCT. In Fig. 4, we see the comparison of each model's accuracy vs the number of samples per class. We show how each model performs when given only 500, 1000, 2000, 3000, 4000, or 5000 (original) samples per class, meaning the total training set ranges from one tenth the size to full. It can be observed that CCT is more robust since it is able to obtain higher accuracy with a lower number of samples per class, especially in the low sample regime.

### 4.5  Performance vs Dimensionality

In order to determine whether transformers are dependant on high dimensional data, as opposed to the number of samples, we experimented with downsampled and upsampled versions of CIFAR-10. In Fig. 5, we present the image dimensionality vs the performance of CCT vsViT-Lite. Both models were trained with images of sizes ranging from 16×16 to 64×64. It can be observed that CCT performs better on all image sizes, with a widening difference as the number of pixels increases. From this, it can be inferred that CCT is able to better utilize the information density of an image, while ViT does not see continued performance increases after the standard 32x32 size.

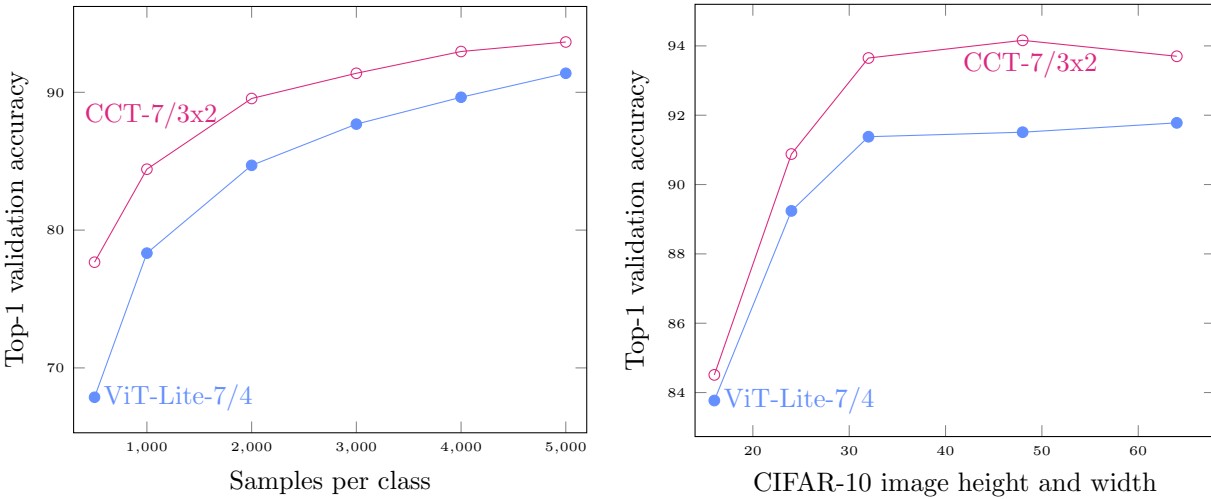

Figure 4: Reduced # samples / class (CIFAR-10)          Figure 5: Image Size vs Accuracy (CIFAR-10)

## 5  Conclusion

Transformers have commonly been perceived to be only applicable to larger-scale or medium-scale training. While their scalability is undeniable, we have shown within this paper that with proper configuration, a transformer can be successfully used in small data regimes as well, and outperform convolutional models of equivalent, and even larger, sizes. Our method is simple, flexible in size, and the smallest of our variants can be easily loaded on even a minimal GPU, or even a CPU. While part of research has been focused on large-scale models and datasets, we focus on smaller scales in which there is still much research to be done in data efficiency. We show that CCT can outperform other transformer based models on small datasets while also having a significant reduction in computational costs and memory constraints. This work demonstrates that transformers do not require vast computational resources and can allow for their applications in even the most modest of settings. This type of research is important to many scientific domains where data is far more limited that the conventional machine learning datasets which are used in general research. Continuing research in this direction will help open research up to more people and domains, extending machine learning research.

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
