# OpenReview forum: "Escaping the Big Data Paradigm with Compact Transformers"
_TMLR — Rejected by TMLR_

### Review · Reviewer_cpmx · 2022-12-04

**Summary Of Contributions:**

This paper proposed compact Vision Transformers (ViT) to achieve good direct-training performance on small and medium size vision classification datasets. There are two major contributions from the paper:

1. Proposed new techniques for better adapting Transformers to vision tasks with limited training data. These techniques include utilizing small patch sizes as inputs, SeqPool with attention-based average pooling for classification, and convolutional tokenizers.

2. Perform experiments on small and medium size datasets to demonstrate the effectiveness of the proposed techniques on training data efficiency. Generate SoA performance on CIFAR-10 for ViT, which is even higher than many convolutional architectures.

**Audience:**

Yes

**Broader Impact Concerns:**

There are no broader impact concerns.

**Claims And Evidence:**

No

**Requested Changes:**

1. Suppose the authors want to show the connection between small-size models and data-efficient training (see Weakness 1). In that case, the results of ViT-12/16 with small patches and convolutional tokenizers need to be added to Table 1. In addition, it's also preferred to add ViT-12/16 with all techniques introduced in Table 7.

2. The authors need to add a section or paragraph discussing the connection between the selected datasets and the medical and scientific tasks that motivated the work (see Weakness 2). Additionally, it's also worth discussing how the proposed work can be generalized to these tasks.

3. In addition to DeiT, the authors must compare the proposed method with other ViT approaches with inductive biases from convolutional networks.

4. It will be more clear for readers if the authors can add the training details in the experiment section.

5. The performance vs. dataset size (C.2 in the appendix) is an important result related to data-efficient training. It's better to show it in the main text and explore more in this direction.

6. What are the parameters for the pooling layers in the convolutional tokenizer?

7. Please separate index numbers for the main text and appendix table and figures.

**Strengths And Weaknesses:**

Strength

The paper shows that introducing data-specific inductive bias to ViT is very important and effective for increasing ViT training accuracy on vision datasets with limited training data. From the experiments, the authors show two techniques that can introduce inductive bias to the ViT and improve performance. First, by utilizing small patch sizes for low-dimensional inputs, and second, by directly using convolutional layers as a tokenizer. Both approaches increase classification accuracies on small datasets compared with the original ViT. The convolutional tokenizers can even outperform many convolutional architectures.

Weakness

1. The primary motivation and claim of the paper need to be clarified and better supported by experiment results. The paper has two different claims: data-efficient training and computation efficiency. However, the experiments fail to support both claims simultaneously, and the primary motivation for data-efficient training cannot be satisfied. For example, the paper hints there is a link between model size and data-efficient training and shows results comparing ViT and Vit-Lite (Table 1, and Table 7 in the appendix). Although ViT-Lite generates better accuracies, the results only prove patch sizes are essential for generating good accuracies on low-resolution datasets. Without additional experiments on ViT with small patches and convolutional tokenizers, the paper can only claim computation efficiency as the advantage of the compact ViT design, but not data-efficient training.

2. The motivation for developing pre-train-free and data-efficient training for medical and scientific domains needs to be better supported by the paper's choice of datasets in experiments. The datasets chosen by the authors work well with pre-trained models and don't reflect the difficulties of medical and scientific datasets, such as low signal-to-noise ratio, unique features required by special sensors, and imbalanced classes, to name a few.

3. The paper lacks comparisons with other ViT architectures that introduce inductive bias of convolutional neural networks into ViT. The experiments from the paper show the convolutional tokenizer benefits data-efficient training. It's interesting to see that such a simple technique works well, but comparison with similar methods is lacking. Many other ViT variances also adopted convolutional inductive biases, such as Swin Transformer [1] using local self-attention inspired by convolution operations, PiT [2] using pyramidal feature maps similar to convolutional neural networks. It will be interesting to see how the proposed architecture compares with these works.

[1] Liu, Ze, et al. "Swin transformer: Hierarchical vision transformer using shifted windows." Proceedings of the IEEE/CVF International Conference on Computer Vision. 2021.

[2] Heo, Byeongho, et al. "Rethinking spatial dimensions of vision transformers." Proceedings of the IEEE/CVF International Conference on Computer Vision. 2021.

---

> ### Author Response · Authors · 2023-02-25
> **Response to reviewer cpmx**
>
> Thank you for your time and feedback.
>
> Below is our response to requested changes:
>
> 1. Thank you for pointing this out, we have added those experiments:
>     * ViT-12/16 with augmentations and tuning
>     * ViT-12/8 and ViT-12/4
>     * ViT-7x1 and ViT-3x2
> 2. Data scarcity is common in medical studies and science in general, and in addition to that, the distribution and domain of
>    of their visual data is quite different from typical computer vision benchmark datasets such as ImageNet, JFT-300M, and the
>    like. The original ViT's key finding was that large scale pretraining is what allows ViTs to exceed CNNs in image
>    classification. We however note that the pretraining will be less useful to cases such as medical studies for that reason.
>    As a result, we primarily focus on training from scratch, and training on datasets with limited number of samples (CIFAR,
>    MNIST, Flowers), and divide them into small-resolution (CIFAR and MNIST) and medium-resolution (Flowers) datasets, both of
>    which are common in the sciences.
>    We also add that Flowers is not only limited in samples; the sample are also highly diverse and belong to one of 102
>    categories, some of which are very difficult to distinguish (also similar to medical data).
>    We also note that our work (as a preprint) has been used in the sciences and medical studies, and cited over 100 times total.
> 3. Please refer to our general response.
> 4. Thank you for bringing this to our attention. We added those details to section 4.
> 5. We have moved those experiments, along with our ablation table, to the main text.
> 6. We used max pooling, and it does not have any parameters.
> 7. Thank you for bringing this to our attention. We have resolved that issue.

---

### Review · Reviewer_2NMB · 2022-12-07

**Summary Of Contributions:**

This paper studies transformer-based algorithms for small-to-medium size datasets in computer vision. The authors start by extending Vision Transformer and make several innovative changes. The proposed models, especially the most comprehensive one (the Compact Convolutional Transformers, CCT), can perform similarly or even better on some benchmark small-to-medium size data sets (e.g. CIFAR-10/100, FLOWERS-102) with much less parameters, compared to start-of-the-art methods.

**Audience:**

Yes

**Broader Impact Concerns:**

Not applied.

**Claims And Evidence:**

Yes

**Requested Changes:**

As mentioned in the weakness part. I hope the authors can:

1. Add theoretical insights behind all of the 3 changes. Also, it will be better if the authors can finish more thorough experiments regarding the effects of adding these changes.

2. Add more details to the baseline models and the metrics.

3. Extend the experiments to other tasks (e.g. regression).

4. Perform confidence interval analysis for the metric values.

**Strengths And Weaknesses:**

Strengths:

1. The proposed models are very strong. For example, It is promising that CCT can reach an accuracy of 98.00% on CIFAR-10 with only 3.76M parameters, whereas the best baseline in the comparison uses 5.70M parameters to reach an accuracy of 97.92%. I think the models are very useful.

2. The authors did very thorough ablation study on the proposed ideas.

3. The paper is well-written and easy to follow.


Weaknesses:

1. The techniques that was applied are not innovative. From my understanding, the major contributions of this paper come from 3 changes of existing method: reducing sizes inside Vision Transformer (Change A), adding a softmax function to the sequence outputs as weights to outputs themselves as a pooling method (Change B), and, applying max pooling + conv2d as a tokenizer (Change C). These are not innovative ideas. Definitely, it is good if we can achieve a good performance with small changes. But it will be better if the authors can better explain the insights behind each of these changes theoretically. Also, it will be better if the authors can test whether all of these 3 changes are necessary. From my understanding, the authors mainly tested ViT, ViT + Change A, ViT + Change A/B, and, ViT + Change A/B/C. It will be better if the authors can try all $2^3=8$ combinations to see whether it is the case that all of them are useful.

2. Some part of the papers might need more details. For instance, the proposed models are based on ViT. It will be great if the author can provide more details on it. In addition, it is unclear how are the MACs computed in all tables. It will be better if the authors can put more details there.

3. The experiments mostly focused on classification. What about other tasks (e.g. regression)?

4. All the accuracy values in the tables are just single numbers. It will be better if confidence intervals can be shown.

---

> ### Author Response · Authors · 2023-02-25
> **Response to reviewer 2NMB**
>
> Thank you for your time and feedback.
>
> Below is our response to requested changes:
>
> 1. As mentioned in our General Response, the common notion, and the claim in the ViT paper suggested the Transformer
>    Encoder's lack of locality and other similar inductive biases limits it from generalizability in small scale datasets. Our
>    approach to studying this problem was asking the following questions:
>     * ViT followed existing large-scale language models, and at the time very few Transformers with limited number of
>           parameters had been studied. In the original paper itself, they scaled datasets such as CIFAR from 32x32 images to
>           224x224 and fine-tuned models large enough to achieve state-of-the-art scores on ImageNet on CIFAR. But what was
>           really lacking was choosing a model with a size appropriate for datasets as small as CIFAR. (Our study began in early
>           2021 and our work has been available as a preprint since then, and has accrued more than 150 citations across fields,
>           but unfortunately received biased reviews leading to its remaining as a preprint until now.)
>           We observed that smaller scale models converge more easily without any additional augmentations introduced.
>     * We also noticed that the default classifier used in ViT, and other Transformer-based classifiers, uses a learnable
>           token that attends to input tokens throughout the model, aggregating context, and then finally used for
>           classification. Because we limited the model size, it would aggregate less context, and therefore possibly limit
>           generalization. As a result, we introduced our sequence pooling method that potentially improves generalization in
>           such depth-limited Transformer Encoders because it pools over output tokens and aggregates context immediately before
>           classification. We found this to be the case in our experiments as well.
>     * Even with an appropriate sized model, ViT still uses a very limited tokenizer to create the *language* that the
>           Transformer learns. While this tokenizer can generalize well when trained on significant sums of data, a more relaxed
>           tokenizer with overlapping convolutions (i.e. our tokenizer) could ease convergence for limited data. We observed that
>           to be true in our experiments.
>
> As for studies showing the effects of each change, we already present that in our ablation study, which we moved from the
>    appendix to the main text.
>    We added experiments that would address your concerns to that table as well:
>    * None: ViT-12/16
>    * A: ViT-Lite
>    * A+B: CVT
>    * A+B+C: CCT
>    * B: ViT-12/16 + SP (= CVT-12/16) (new)
>    * C: ViT-12/7x1 and ViT-12/3x2 + CT (new)
>    * B+C: ViT-12/7x1 and ViT-12/3x2 + SP (=CCT-12) (new)
>    * A+C: CCT-7/7x1 and CCT-7/3x2 + CT (new)
> 2. MACs (used interchangeably with FLOPs in many papers) are the total number of floating point multiplies and additions, and
>    are a standard metric for expressing the number of computations required for an input of a specific size.
>    We computed MACs for all experiments that we ran using a standard pytorch extension, and all of these details will be open
>    sourced with our code.
> 3. Yes, ViT was a model intended for image classification. Since our focus was directly addressing ViTs' generalizability in
>    data-limited domains, we focused on image classification as well.
>    That said, the model can be extended to other tasks, but it was not our focus.
> 4. Unfortunately we did not iterate most of our experiments and are therefore unable to present confidence intervals without
>    rerunning all of our experiments multiple times for that purpose.

---

### Review · Reviewer_WJYh · 2023-01-14

**Summary Of Contributions:**

This work presents Compact vision transformers and compact convolutional transformers to extend transformer-based approaches to small datasets. The authors introduce a sequence pooling strategy and convolution-based tokenization to improve the performance of ViT-like architectures and report good results on various datasets.

**Audience:**

Yes

**Claims And Evidence:**

No

**Requested Changes:**

Please compare with other works which aim to also make ViTs lightweight as mentioned above.

**Strengths And Weaknesses:**

Strengths
1. The results on small datasets are impressive
2. The sequence pooling strategy seems to work better than non-parametric approaches like average pooling
3. Convolution-based feature extraction in tokenization improves performance

Weaknesses
There is a massive corpus of recently proposed lightweight vision transformer papers that the authors do not compare to which makes it difficult to gauge the results obtained. Since this paper is relatively limited in novelty with other authors [Heo et al.] also proposing similar ideas, I would like to see a more thorough comparison with at least the following works:
1. Park and Kim. How do vision transformers work?
2. Wang et al. Pyramid vision transformers
3. Heo et al. Rethinking Spatial Dimensions of Vision Transformers
4. Liu et al. Swin Transformer
5. D'ascoli et al.  ConViT

This should be straight-forward to carry out as the datasets in question are small as the authors suggest. The authors also do not mention if the other methods use the same augmentation strategy or not.

---

> ### Author Response · Authors · 2023-02-25
> **Response to reviewer WJYh**
>
> Thank you for your time and feedback.
> We've added the works referenced to section 2 (excluding those already cited.)
> Please refer to our General Response, in which we outline the reasons why we did not compare to hierarchical vision
> transformers and hybrid models, and limited our study to the original ViT family.
>
> As for augmentations, we use the same augmentations and training techniques adopted by the community (through the timm library),
> as well as works such as DeiT, Tokens-To-Token ViT, and the like.
> To our knowledge, every method since ViT has adopted these settings when training on image classification.
> We've revised section 4 and clarified training settings.

---

### Author Response · Authors · 2023-02-25
**General Response**

We thank the reviewers and chairs for their valuable time and feedback.
We apologize for the delay in our response.

Herein we address a common concern raised by reviewers regarding the baselines we compare our method to.
The original Vision Transformer is a model with very limited inductive biases, as pointed out in the original ViT paper:

_"Transformers lack some of the inductive biases inherent to CNNs, such as translation equivariance and locality, and therefore do not generalize well when trained on insufficient amounts of data."_

ViT authors stated that the observed lack of generalizability is probably due to the lack of inductive
biases present in CNNs.
The focus of our paper was to study whether or not this is a limitation of the model (the Transformer Encoder), the
tokenizer, the classifier, or the training techniques.
The common notion, and the claim in the ViT paper suggested it was the Transformer Encoder.
To investigate, we _had to_ keep the original Transformer Encoder unchanged, and show how well they can truly generalize.
Our approach consisted of: adapting model size and patch size to the problem (almost impossible to do and maintain
pre-training), creating a better *classifier* head by pooling over tokens with attention instead of relying on the class token (better post-processing),
and creating a better *language* by introducing a convolutional tokenizer (better preprocessing).
In doing so, we still maintain the same vanilla Transformer Encoder, while that does not hold true for hybrid and
hierarchical models that directly introduce convolutions, or modules with similar inductive biases, into the model at the
layer-level.


It is true that many works were inspired by Vision Transformer, and introduced local inductive biases into the Transformer
encoder layers.
We have added the referenced works (PVT, Swin Transformer, PiT), and were already citing some of them (ConViT).
However, we do not believe that comparing to those models is relevant to the study in the paper because:
1. Works such as PVT, Swin Transformer, ConViT, and PiT introduce modules with those inductive biases in *every layer*, while our
   work focuses on the minimum level of changes to the original ViT model that would make it directly applicable to
   low-data/low-dimensionality tasks. ViT-Lite is identical in architecture to ViT; CVT adds our sequence pooling module (which
   is directed at improving classification convergence and not introducing local biases); and CCT replaces the existing
   convolutional tokenizer with a slightly different one (two convolutional layers with pooling
   instead of a single non-overlapping convolution.)
   This is how our work differs from those referenced; their focus was to create a new branch of models (hierarchical vision
   transformers) whose primary focus is applicability to downstream tasks.
2. This paper is not focused on proposing a new architecture, unlike the referenced works. It is simply investigating a strong
   claim made in the original ViT paper (Transformers lack some of the inductive biases inherent to CNNs ... and therefore do
   not generalize well when trained on insufficient amounts of data.) We do so by maintaining the same Transformer architecture,
   and instead improving the classification and tokenization, along with training techniques, and observe that those play a
   crucial role in the model's performance. We show that a pure Transformer encoder, with our proposed improvements to the pre-
   and post-processing modules, can compete with CNNs in data-limited settings.


We would also add that our work was originally done in early 2021, concurrent to many of the referenced works.
While we recognize the importance of citing those methods in our Related Works as follow ups to ViT,
we feel that studying them in our work presents a significant amount of effort that would shift the focus of our paper.

We would also add that when we refer to tokenization in the paper, we are in fact referencing both tokenization and embedding (patch + embed).
We would be happy to change the use of this phrasing in the paper, particularly in figure 2 if the reviewers agree it clarifies the message.

We thank you again for your time and feedback, and we look forward to your response to this revision.

---

### Decision · Action_Editors · 2023-03-17

**Recommendation:** Reject

**Comment:**

I read through the belated author rebuttal and I am not fully convinced that it addresses the reviewer concerns. While I understand that this was work done in 2021, ~2 years have passed since then, and therefore it's completely reasonable to ask the authors to compare with the other advances in the field ever since.

**Audience:**

I believe the findings of this work, especially if properly situated, would be useful to a relatively large number of people in TMLR's audience, especially those interested in training transformer-like architectures in a small data regime.

**Claims And Evidence:**

This work presents compact vision transfomers and compact convolutional transformers and analyzes them in the small data regime. A new sequence pooling strategy and tokenization has showed good results on small datasets. All the reviewers commented that the results are strong and that the proposed technique seems sound.

On the other hand, all three reviewers remain unconvinced by the comparisons with existing literature that proposes similar ideas. There are plenty of works out there that introduce various inductive biases which make transformers work in a small data regime, many of them pointed to by the reviewers. I understand the fine nuances of layerwise vs. model-wise inductive biases that the authors are talking about in their rebuttal, but I have to agree with the reviewers that it's ultimately these comparisons should be made. This is because they fundamentally explore very similar corners of the conceptual space and this work would be stronger if it was situated and explained so.

Given the focus on small data regime, I don't believe doing these comparisons would be onerous. I think this work would be significantly stronger if the comparisons were there and discussed.